# Cervical Spine Pedicle Screw Accuracy in Fluoroscopic, Navigated and Template Guided Systems—A Systematic Review

**Arin Mahmoud \*, Kanatheepan Shanmuganathan, Brett Rocos 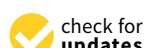, Fady Sedra, Alexander Montgomery and Syed Aftab**

Trauma and Orthopaedic Department, Royal London Hospital, London E1 1FR, UK; kanatheepan.shanmuganathan@nhs.net (K.S.); brett.rocos@nhs.net (B.R.); f.sedra@nhs.net (F.S.); alexander.montgomery@nhs.net (A.M.); syed.aftab@nhs.net (S.A.)
\* Correspondence: arin.mahmoud1@nhs.net

**Abstract:** *Background:* Pedicle screws provide excellent fixation for a wide range of indications. However, their adoption in the cervical spine has been slower than in the thoracic and lumbar spine, which is largely due to the smaller pedicle sizes and the proximity to the neurovascular structures in the neck. In recent years, technology has been developed to improve the accuracy and thereby the safety of cervical pedicle screw placement over traditional fluoroscopic techniques, including intraoperative 3D navigation, computer-assisted Systems and 3D template moulds. We have performed a systematic review into the accuracy rates of the various systems. *Methods:* The PubMed and Cochrane Library databases were searched for eligible papers; 9 valid papers involving 1427 screws were found. *Results:* fluoroscopic methods achieved an 80.6% accuracy and navigation methods produced 91.4% and 96.7% accuracy for templates. *Conclusion:* Navigation methods are significantly more accurate than fluoroscopy, they reduce radiation exposure to the surgical team, and improvements in technology are speeding up operating times. Significantly superior results for templates over fluoroscopy and navigation are complemented by reduced radiation exposure to patient and surgeon; however, the technology requires a more invasive approach, prolonged pre-operative planning and the development of an infrastructure to allow for their rapid production and delivery. We affirm the superiority of navigation over other methods for providing the most accurate and the safest cervical pedicle screw instrumentation, as it is more accurate than fluoroscopy and lacks the limitations of templates.

**Keywords:** cervical spine; pedicle screw

## 1. Introduction

Complications from cervical spine surgery have the potential for catastrophic complications because of the proximity to the vertebral arteries, the airway and the spinal cord. Furthermore, constructs applied to the cervical spine may be less robust because of the smaller size of the anchors and the minimal amount of bone available for purchase.

Pedicle screws provide excellent biomechanical stability throughout the spine; however, its use in the cervical spine is limited by the technical challenges of instrumenting the narrow, oblique pedicles, and the risk of damage to the vertebral arteries [1–4]. To overcome these challenges, Abumi demonstrated the safe use of cervical pedicle screws inserted using a combination of freehand and lateral fluoroscopic guidance, which has become the standard for cervical pedicle screw placement, but this is not without technical hurdles [5–7]. Despite familiarity with the technique, the incidence of neurovascular complications due to screw placement with fluoroscopic guidance in the cervical spine has been estimated to be as low as 0.4%, although the consequences of these infrequent events can lead to paraplegia, cerebral infarct and death in severe cases [1,8,9].

A solution to the difficulties in reliably instrumenting the cervical pedicle could be technological; namely, navigated surgery. Developments in intra-operative 3-dimensional (3D) image acquisition now permit real-time 3D models of the spine and the relative position of instruments to be displayed on a monitor that a surgeon can use to accurately place screws without the benefit of direct vision. Previous studies have suggested that these technologies not only improve the safety of pedicle screw placement, but also reduce both the duration of surgery and the radiation exposure to the surgical team from repeated fluoroscopic imaging [7,10,11]. Several variations in navigation technology exist. Systems may utilise preoperative scans or CT or MRI scans to display the anatomy relative to a reference frame attached to a fixed bony point on the patient, creating a virtual 3D model of the instruments relative to the bone [7,12,13].

More recently, 3D printing has advanced and has found application in medicine; pre-operative imaging creates computerised models of the vertebrae, and 3D templates are designed which fit onto the posterior surface of the vertebrae, with screw guides that guide the direction of the screws along the trajectory of the pedicle. This negates the need for intraoperative imaging; however, the bespoke nature of the templates means time is needed to model, print and distribute them and it is not appropriate for use in cases where there is no lead time (such as in emergent surgery) [4,6,14,15].

The choice of the navigating technique depends upon the training and preference of the surgeon, the availability of the required technology and the clinical urgency. The relative success of each technique, as evaluated by pedicle breach rates (and thereby accuracy and clinical outcomes) and reduction in radiation exposure is, as yet, ill-defined because of the rapid pace of technology and the variations in experience, preference and availability of each system. To overcome this challenge, this article reports the outcome of a systematic review of the relative accuracy of fluoroscopic-guided, computer-navigated and 3D template techniques when employed to place pedicle screws in the cervical spine.

## 2. Methods

A structured literature search was employed using the PubMed database and the Cochrane library. Reference snowballing was used to ensure a comprehensive search, and expert opinion leaders were consulted for any additional resources. Databases were searched using the keywords 'Pedicle Screw' and 'Cervical Spine'.

Studies were included if they were randomised controlled or observational trials which compared one or more of the fluoroscopy, computer-navigated, or 3D-mold techniques when placing pedicle screws in the cervical spine on humans in vivo and if they measured accuracy with post-operative CT scanning. No restrictions were placed on the underlying pathology, the spinal levels instrumented, age, gender, ethnicity or geography.

## 3. Results

The search and selection process can be summarised in the PRISMA Diagram displayed in Table 1. From a total of 50 papers in the search, 9 were deemed appropriate for inclusion in the synthesis.

Post-operative CT scans were used to determine the screw accuracy and various measures of accuracy were used—we amalgamated these into simply measuring whether the cortex was breached or not. Scheuffler's trial measured the accuracy of a computer-assisted programme by how many millimetres the pedicle screw had deviated by. We included a deviation of >2 mm as equivalent to a cortical breach [10]. Studies of cervical anatomy in the White American population have shown that this is appropriate: 75.5% of C3 pedicles had a diameter <4 mm. A deviation of 2 mm in any direction in these cases may breach the cortex [16].

Different navigation systems were used, including StealthStation [12,17], VectorVision [18,19] and intraoperative CT [10,20]; both were included. The trials involving templates used Mimics Software to design the template from pre-operative scans. Three-dimensional printing was used to create physical copies [14,15].

**Table 1.** PRISMA diagram outlining how the systematic review was conducted.

| **Pedicle Screw Accuracy in the Cervical Spine- A Systematic Review** | |
| --- | --- |
| PubMed Data range searched- Pedicle Screw Cervical Spine; Clinical Trials and Randomised Control Trials Results—27 | Cochrane Library Data range searched- Pedicle Screw Cervical Spine; Trials Results—29 |
| ↓ | ↓ |
| Total number screened following exclusion of duplicates—50 | |
| ↓ | ↓ |
| Articles undergoing full analysis—14 | Obviously irrelevant, inappropriate articles excluded following review of abstract—36 |
| ↓ | |
| Inclusion Criteria- in vivo human trials; pedicle screw placement in the cervical spine; method of guidance detailed; post operative CT scanning to determine accuracy All aetiologies, all vertebrae in the cervical spine, all languages | |
| Total Number of Papers Included—9 | Excluded—5 |

A total of 9 valid papers were identified, comparing accuracy in a total of 1427 screws across the different methods.

The results are summarised in Tables 2 and 3. Table 2 shows the results broken down by paper and method. Table 3 shows the results for each method across the various trials, and shows that the accuracy rates in relation to fluoroscopic methods were 80.6%, 91.4% in relation to the navigation methods and pedicle screws inserted through templates had an accuracy rate of 96.7%.

**Table 2.** Summary of all the trials included in the systematic review and the results.

| | | **Total** | **Accurate** | **Cortex Breached** |
| --- | --- | --- | --- | --- |
| Yukawa 2006 [21] | Fluoroscopy | 419 | 359 | 60 |
| Kotani 2003 [12] | Navigation | 78 | 77 | 1 |
| Ito 2006 [17] | Navigation | 25 | 20 | 5 |
| | Fluoroscopy | 86 | 26 | 60 |
| Richter 2004 [19] | Navigation | 31 | 30 | 1 |
| Jiang 2017 [15] | Template | 100 | 96 | 4 |
| | Fluoroscopy | 116 | 103 | 13 |
| Scheuffler 2010 [10] | Navigation | 138 | 117 | 21 |
| Rajan 2010 [18] | Navigation | 98 | 89 | 9 |
| Yu 2017 [20] | Fluoroscopy | 132 | 117 | 15 |
| | Navigation | 128 | 122 | 6 |
| Xiong 2017 [14] | Fluoroscopy | 24 | 21 | 3 |
| | Template | 52 | 51 | 1 |

**Table 3.** Synthesis of all trials according to method of pedicle screw placement.

|  | Total | Accurate | Cortex Breached | Percentage |
|---|---|---|---|---|
| Fluoroscopy | 777 | 626 | 151 | 80.57 |
| Navigation | 498 | 455 | 43 | 91.37 |
| Template | 152 | 147 | 5 | 96.71 |

Navigation methods provide improved accuracy to fluoroscopy; however, the novel method of template guidance had the highest accuracy rate, although screws inserted this way accounted for 10.7% of the total.

Chi squared tests were used to calculate the significance of the results, displayed in Tables 4 and 5. The statistical analysis found that navigation is significantly more accurate than fluoroscopy, while templates are significantly more accurate than navigation.

**Table 4.** Navigation methods are significantly more accurate than fluoroscopy. The chi-squared statistic is 27.4356. The *p*-value is <0.000001.

|  | Accurate | Inaccurate | Marginal Row Totals |
|---|---|---|---|
| Fluoroscopy | 626 (658.77) [1.63] | 151 (118.23) [9.09] | 777 |
| Navigation | 455 (422.23) [2.54] | 43 (75.77) [14.18] | 498 |
| Marginal Column Totals | 1081 | 194 | 1275 (Grand Total) |

**Table 5.** Templates are significantly more accurate than Navigation methods. The chi-square statistic is 4.8647. The *p*-value is 0.027412.

|  | Accurate | Inaccurate | Marginal Row Totals |
|---|---|---|---|
| Navigation | 455 (461.22) [0.08] | 43 (36.78) [1.05] | 498 |
| Template | 147 (140.78) [0.28] | 5 (11.22) [3.45] | 152 |
| Marginal Column Totals | 602 | 48 | 650 (Grand Total) |

## 4. Discussion

Pedicle screw insertion in the cervical spine is complicated by the smaller pedicle widths, the larger transverse pedicle angles and the proximity of the vertebral arteries, nerve roots and spinal cord. Factors affecting anatomy include gender, age, height and race [6]. Figure 1 shows a post-operative axial CT image of a patient who underwent cervical spine pedicle screw fixation with a fatal injury to the vertebral artery. Several authors have converged on the 4 mm pedicle width as the smallest sized pedicle that would be appropriate for pedicle screws [5,6,22]. The thickness of soft tissues over the posterior neck must also be considered—thick muscle and fat has been suggested to be a greater cause of pedicle wall violation than pedicle diameter [6].

Lateral mass screw fixation has been used in the cervical spine since 1964, three decades before Abumi demonstrated the use of pedicle screws [23]. The screw trajectory in lateral mass screws means significant neurovascular complications are less likely, although lateral mass fractures can take place [2,23]. However, numerous authors comparing lateral mass and pedicle screw fixation have concluded pedicle screws provide stronger fixation and higher pull-out strengths [6,23–25]. Lateral mass screws remain useful in most clinical circumstances including in patients with anomalous vertebral artery anatomy [6] or with single level ligamentous instability [2]; however, the superior biomechanical properties of pedicle screws mean their popularity is increasing for use in the cervical spine.

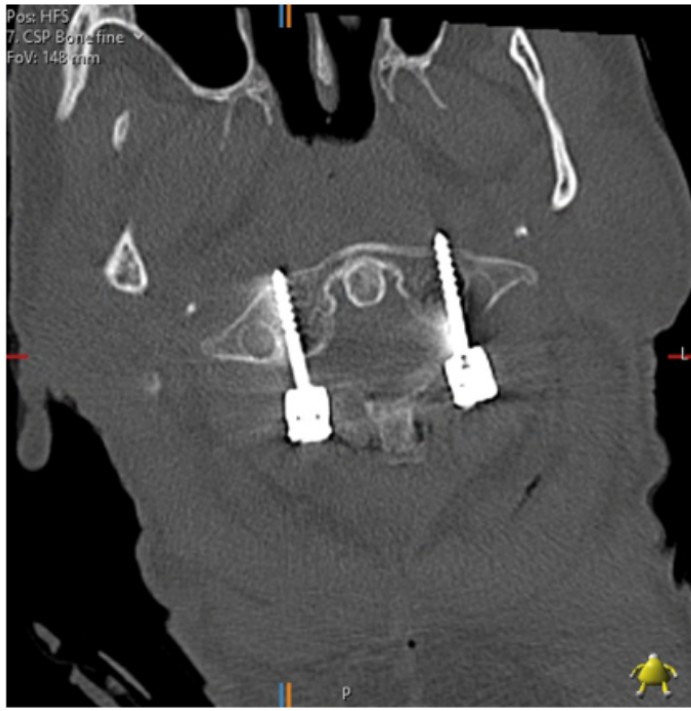

**Figure 1.** C1/2 fixation performed under fluoroscopic guidance; the left foramen transversarium has been penetrated by a pedicle screw. The patient suffered a massive posterior stroke and died within 24 h.

Small cortical breaches may not produce neurovascular injury or any worse clinical outcomes; however, there is no consensus in the literature regarding the dimensions of a safe zone. Therefore the aim at present should be to avoid any cortical breach [3], which requires absolute accuracy in view of the sizes of the structures being instrumented.

Freehand fluoroscopic techniques remain widely used, but technological advancements in CT scanners and computing have permitted new techniques which provide a higher accuracy throughout the spine. Two broad methods of navigation currently exist: intraoperative CT scanning providing real-time 3D navigation and preoperative CT registration in relation to a fixed reference with intraoperative computer assistance. Potential causes of error in Computer Assisted systems include reference clamp movement due to vibration from drilling or difficulty of secure fixation to the vertebrae due to their small size [2]. These techniques are not clearly defined and separate and have many variations, overlaps and systems produced by different manufacturers.

Large meta-analyses by Verma [26] involving 5992 pedicle screws found improved accuracy in computer-assisted techniques compared with non-navigated techniques; similarly, Shin found improved accuracy with intraoperative CT navigation over 4814 navigated and 3725 non-navigated screws [27]. Gelalis noted in a systematic review of over 6617 pedicle screws that CT navigation is more accurate than fluoroscopy and freehand techniques [28]. While these studies provide evidence that navigation is more accurate, they include thoracic, lumbar and sacral regions. Results from this systematic review suggest that navigated techniques provide superior accuracy to fluoroscopy in the cervical spine, in line with studies on other parts of the spine.

Although accuracy is crucial, other aspects of these techniques need to be considered.

When evaluating this approach, radiation exposure to the patient, surgeon and operating room personnel should be considered. Bratschitsch compared radiation exposure in 37 patients who had either fluoroscopy or navigation using an intraoperative CT—they found that intraoperative CT navigation caused more radiation exposure to the patient but less radiation exposure to the staff [29]. Navigation using preoperative registration greatly

reduced the use of an intraoperative CT, and thus radiation exposure to staff, although patients will require a pre-operative CT [30].

While in principle, preoperative registration may eliminate the need for intraoperative scanning, the exceptional mobility of the cervical spine and the fact that pre-operative images are taken with the patient supine whilst the operation is performed prone, may cause changes to the relative positions of the reference clamp and vertebrae and recalibrate the model. Furthermore, the cervical spine is usually a highly mobile structure and it is difficult to intra-operatively mimic the exact position in which the vertebrae were scanned pre-operatively. Intra-operative registration accuracy may be recalibrated by micromovements from breathing and instrumentation. Therefore Richter advises the availability of intraoperative fluoroscopy or CT navigation for verification through intraoperative imaging or as a reserve if the navigation system crashes [2].

Aspects of time may be divided into pre-operative, intra-operative and post-operative. Navigation methods involving pre-operative registration require time to implant the reference frame, scan and model the patient; fluoroscopy and intraoperative CT navigation ideally would involve pre-operative scanning for planning, but do not require it, and can thus be used in emergent situations.

Pre-operative registration navigation was previously held back by prolonged intraoperative registration that had to be performed for each vertebrae, extending intraoperative times. Rajan reported on over 340 pedicle screws and showed that average insertion time was 2.37 min, whilst the registration time was 24.6 min [18]. Yang compared 210 pre-operatively registered and navigated pedicle screws with 152 fluoroscopic lumbar spine pedicle screws and noted 3.65 min per screw in the navigated group and 4.43 min in the fluoroscopy. He also stated that newer models have reduced registration times to 14–18 min [31].

Several studies have shown intraoperative CT navigation to prolong operative times-Meng's meta-analysis involving 9019 thoracic screws found that although CT navigation improved accuracy, it significantly prolonged operative time [32]. Similarly, Mirza compared fluoroscopy, intraoperative CT and pre-operative registration navigation techniques on thoracic cadavers, noting that standard fluoroscopy was significantly faster [33]. Kotani compared 222 intraoperative CT navigations with 416 pre-operatively registered thoracic pedicle screws, noting that intraoperative CT navigation was significantly slower [34].

Post-operative recovery time is determined by accuracy, adverse events, operative time and whether open or minimally invasive techniques were used.

Templates are a novel concept in spinal surgery, designed to eliminate inaccuracies that can arise during navigated techniques from intraoperative movements. They are composed of a component designed to sit on the posterior aspect of a vertebrae, contoured to its precise anatomy and a cylindrical guide for drilling. Provided good contact between the template and vertebra can be made and maintained, the cylinders will be in line with the planned pedicle screw trajectory. They are made by 3D printing, or additive manufacturing, where a computerised 3D model of an object is sliced into cross-sectional layers, which is then created by selectively placing the material in successive layers. This method can create complex geometrical structures and functional parts and allows an easy adaptation of designs [35].

The proposed benefits of this method include a reduced intraoperative time—on average, it takes approximately 80 s from fixing the template to the vertebrae to inserting the pedicle screw [6]. The template techniques included in this review required pre-operative CT scanning, as well as intraoperative fluoroscopic verification; however, both studies noted significantly reduced radiation exposure [14,15]. Kaneyama noted in their experience with templates that intraoperative fluoroscopy was unnecessary and even impeditive [4]. However, prolonged pre-operative planning is required for templates—it will require pre-operative scanning and although printing a template only takes approximately 3–4 h, it can take up to a week for the template to be ready for use [4,6].

Another dynamic at play within surgery at large is open versus minimally invasive surgery (MIS); pedicle screw insertion may be completed under either. Although it has not been demonstrated conclusively that MIS provides superior outcomes to open surgery in pedicle screw insertion, there is a general move towards MIS due to lower rates of soft tissue trauma, infection and blood loss and improved cosmesis [36]. Minimally invasive techniques have been demonstrated in fluoroscopy and navigation techniques [36,37]; template guided techniques require open surgery, and the nature of the technique may mean this is unavoidable, as the template requires cleanly dissected posterior vertebrae [4,14,15]. MIS techniques of pedicle screw insertion have not yet been demonstrated in the cervical spine; however, we are certain that navigation techniques will ultimately lead to the development of MIS cervical pedicle screw instrumentation.

## 5. Conclusions

Cervical spine anatomy makes the placement of pedicle screws in this location more difficult and dangerous. Various technologies have been developed to improve accuracy rates.

Our results demonstrate that navigation techniques provide superior accuracy to fluoroscopy; however, they may increase operation times and radiation exposure.

Although templates show a higher accuracy, they have the smallest sample size and their practical application is currently limited by their novelty and prolonged preoperative planning.

**Author Contributions:** Conceptualization, data collection, research, writing manuscript, submission, A.M. (Arin Mahmoud); research, data collection, K.S.; spinal surgery consultant, provided advice and edited the manuscript, B.R.; spinal surgery consultant, provided advice and guidance, F.S.; spinal surgery consultant, provided advice and guidance, A.M. (Alexander Montgomery); conceptualization, edited manuscript, consultant who provided the main supervision and guidance for paper, S.A. All authors have read and agreed to the published version of the manuscript.

**Funding:** This research received no external funding.

**Institutional Review Board Statement:** Not applicable.

**Data Availability Statement:** Articles providing data for systematic review are all referenced in the bibliography.

**Acknowledgments:** Statistical Consultation provided by Laurence Bernstein-Newman, Freelance Data Analyst; laurence@lbn.me.uk.

**Conflicts of Interest:** The authors declare no conflict of interest.

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
