# Peer review of "Cervical Spine Pedicle Screw Accuracy in Fluoroscopic, Navigated and Template Guided Systems—A Systematic Review"

_tomography, doi:10.3390/tomography7040052_

Round 1
Reviewer 1 Report
I realize that authors have many journals to consider when they want to publish their work, so I appreciate your interest in Tomography.
I am very happy to be able to write in a positive way but It may be that you would like to consider improving the manuscript in which case I hope that the comments from my review may help you to revise it before resubmitting it.
These comments are given below. It is evident that you have put a great deal of effort into this project and I want to praise your efforts.
Abstract section:
Q1. The abstract needs to be reworked to increase clarity. My suggestion is that what is presented in the results needs to be mentioned in the methods first – with the possibility to leave the less important outcome measures outside the abstract to meet word count limit.
Q2. Regarding the keywords, please use MeSH-terms.
Methods section:
Q3. Please use a Prisma chart.
Author Response
I appreciate your review and constructive feedback.
I have tried to adjust the abstract, particularly its conclusion, to make everything more succinct and clear.
I have adjusted the keywords as per the mesh headings found here https://meshb.nlm.nih.gov/search
I have made major edits to the introduction, methods and results- much of this was done since the submission by Mr Brett Rocos, who I would like to add as a new author. He had previously advised me to insert a prisma diagram which I have done.
Reviewer 2 Report
General impression
In this article, the authors performed a systemic review into the accuracy of various methods of cervical spine pedicle screw fixation. And they concluded that navigation techniques provide superior accuracy to fluoroscopy, however they may increase operation times and radiation exposure.
I guess the information in this paper must be valuable for the spine surgeons doing cervical spine surgery. For these reasons, I think this manuscript is appropriate for publication.
However, I have a couple of minor requests to be revised as stated below. After they have been resolved, I will judge this manuscript can be accepted and published by the Tomography journal.
- Abstract line 12, 19, 21
“pedicle” should be replaced by “Pedicle". It must be started with a capital letter.
“fluoroscopic” should be replaced by “Fluoroscopic". It must be started with a capital letter.
“superior” should be replaced by “Superior ". It must be started with a capital letter.
- Materials and methods line 98
Is “all aetiologies and cervical vertebrae were included” correct? I am not sure but I think “all aetiologies in cervical vertebrae were included” may be correct.
- Results line 116
Is “19.6%” correct?
I am not sure but I think “10.7” may be correct.
152 / (777+498+152) x 100 = 10.7
- Discussion line 169-175 and 176-183
Mostly same paragraphs were repeated. I guess the former paragraph can be deleted.
Author Response
I appreciate your time in reviewing and providing constructive feedback.
I have made major changes to the manuscript- these were largely done by Mr Brett Rocos, a spinal consultant, who I wish to add as an author. His contribution includes restructing the intro, methods and results and advising me to create a PRISMA diagram.
Regarding your specific comments:
-Abstract has been capitalised where needed
-Wording in materials and methods adjusted
-Results- maths error adjusted (thank you for spotting it)
-Discussion- repetitive line removed